# High Incidence and Early Onset of Urinary Tract Cancers in Patients with BK Polyomavirus Associated Nephropathy

**DOI:** 10.3390/v13030476

**Published:** 2021-03-14

**Authors:** Yi-Jung Li, Hsin-Hsu Wu, Cheng-Hsu Chen, Hsu-Han Wang, Yang-Jen Chiang, Hsiang-Hao Hsu, See-Tong Pang, Robert Y. L. Wang, Ya-Chung Tian

**Affiliations:** 1Kidney Research Center and Department of Nephrology, Linkou Chang Gung Memorial Hospital, Taoyuan 333, Taiwan; r5259@cgmh.org.tw (Y.-J.L.); tomwu38@gmail.com (H.-H.W.); hsianghao@gmail.com (H.-H.H.); 2Department of Medicine, Chang Gung University, Taoyuan 333, Taiwan; seanwang@cgmh.org.tw (H.-H.W.); jacobpang@cgmh.org.tw (S.-T.P.); 3Division of Nephrology, Department of Internal Medicine, Taichung Veterans General Hospital, Taichung 407, Taiwan; cschen920@yahoo.com; 4Department of Urology, Linkou Chang Gung Memorial Hospital, Taoyuan 333, Taiwan; zorro@cgmh.org.tw; 5Department of Biomedical Sciences, College of Medicine, Chang Gung University, Taoyuan 333, Taiwan; yuwang@mail.cgu.edu.tw

**Keywords:** BK polyomavirus, urinary tract cancers, BK polyomavirus-associated nephropathy, kidney transplantation

## Abstract

Over-immunosuppressed kidney transplant recipients are susceptible to malignancies and BK polyomavirus (BKPyV)-associated nephropathy (BKPyVAN). This study aimed to verify the association between BKPyV infection and urinary tract cancers (UTC). A total of 244 kidney transplant recipients were enrolled at Chang Gung Memorial Hospital from June 2000 to February 2020. Biopsy-proven BKPyVAN patients (*n* = 17) had worse kidney function (eGFR: 26 ± 13.7 vs. 47.8 ± 31.0 mL/min/1.73 m^2^). The 5-year allograft survival rates for patients with and without BKPyVAN were 67% and 93%, respectively (*p* = 0.0002), while the 10-year patient survival was not different between the two groups. BKPyVAN patients had a significantly higher incidence of UTC compared to the non-BKPyVAN group (29.4% vs. 6.6%). Kaplan–Meier analysis showed that the UTC-free survival rate was significantly lower in BKPyVAN patients, and the onset of UTC was significantly shorter in BKPyVAN patients (53.4 vs. 108.9 months). The multivariate logistic regression analysis demonstrated that age (RR = 1.062) and BKVAN (RR = 6.459) were the most significant risk factors for the development of UTC. Our study demonstrates that BKPyVAN patients have greater allograft losses, higher incidence, a lower cancer-free survival rate, and an earlier onset with a higher relative risk of developing UTC compared to non-BKPyVAN patients.

## 1. Introduction

Kidney transplant recipients have an immunosuppressed status, and defective immune surveillance and clearance increases the risk of malignancies in this population. The most common cancers in Western countries are skin cancers, non-Hodgkin’s lymphoma, and Hodgkin’s lymphoma [1,2]. In contrast, the most common cancers in Taiwan are urinary tract cancers, especially urothelial carcinoma of the kidney and bladder [3,4,5]. Therefore, determining the possibly different etiologies of urothelial carcinoma in kidney transplant recipients is crucial.

Approximately 10–20% of worldwide cancers are attributed to viral infections, and several viruses are reported to be associated with cancer development, including hepatitis B virus (HBV), hepatitis C virus (HCV), human papillomavirus (HPV), human T-cell lymphotropic virus (HTLV), Epstein-Barr virus (EBV), and human herpes virus 8 (HHV8) [6,7]. Recently, Merkel cell polyomavirus (MCPyV) was discovered. MCPyV is an oncogenic virus important for the development of Merkel cell carcinoma (MCC), a rare but highly aggressive human skin cancer [8]. In kidney transplant recipients, oncogenic virus-related malignancies are significantly increased [9]. Viral oncogenesis contributes to or facilitates cancer development through the expression of oncogenic proteins or virus-induced chronic inflammation [9].

In addition to MCPyV, several members of the polymaviridae family, including simian virus 40 (SV40), polyomavirus JC (JCPyV), and BK polyomavirus (BKPyV) have oncogenic properties in cell culture and animal models. Among these polyomaviruses, the role of SV40 in the onset of human cancers remains to be clarified. Indeed, many investigations reported the association between SV40 and human cancers, while other studies published negative results [10]. BKPyV DNA was discovered in renal cell carcinoma, urothelial carcinoma and prostate cancer, while JCV DNA is mainly found in colorectal and CNS cancers [11,12,13]. However, the direct causal role of these viruses in tumorigenesis remains inconclusive.

An epidemiologic study demonstrated that approximately 80% of the general population was infected by BKPyV in their childhood [14]. Following primary infection, BKPyV is persistently hidden in the renourinary tract, and when the immunity of the organ transplant recipients is suppressed, BKPyV rapidly replicates to cause renal inflammation, the occurrence of BKPy-viruria and BKPy-viremia, and sometimes BKPyV-associated nephropathy (BKPyVAN) [15,16,17,18]. Potent immunosuppressants reduce acute rejection episodes but increase the risk of BKPyVAN and its associated allograft loss. A reduction or modification of immunosuppressants may alleviate the progression of BKPyVAN [19,20,21,22].

Despite our good understanding of the association between BKPyV infection and allograft dysfunction, whether BKPyV can induce human cancers remains controversial. The oncogenicity of BKV was discovered in cell cultures, as the introduction of the BKV large T antigen (TAg) into rodent and human cells facilitated oncogenic transformation of the cells [17,23,24]. However, the direct causal role of these small DNA tumor viruses in human cancers needs to be investigated further with new innovative methods. Several studies only show indirect evidence, with the existence of BKPyV DNA or TAg proteins highlighted in the tissue specimens of several human cancers [25,26,27]. Kenan et al. recently reported that the genome of a novel BKPyV CH-1 strain was successfully integrated into the human genome in a urothelial carcinoma specimen and a renal cell carcinoma specimen, indicating the oncogenic potential of BKPyV in humans [28,29].

We previously reported post-kidney transplant urothelial carcinoma of the urinary bladder and renal pelvis in association with BKPyV infection [30]. Most studies reporting an association between urinary tract cancers and BKPyV infection are case-report studies [31,32]. Only a few papers used a cohort study to assess this association but with a relatively short follow-up among BKPyVAN patients [33,34]. The most common cancer in Taiwanese kidney transplant recipients is urinary tract cancer. We, therefore, analyzed the risk of urinary tract cancers among patients with and without biopsy-proven BKPyVAN in a 20-year cohort.

## 2. Materials and Methods

### 2.1. Patients and Definitions

This cohort observational study included 244 kidney transplant recipients at Chang Gung Memorial Hospital from June 2000 to February 2020. Ethics approval (201801596B0) was obtained from the Medial Ethics Committee of Linkou Chang Gung Memorial Hospital. These recipients received regular screening for BKPy-viruria and viremia. If the urinary BKPyV load was more than 1 × 10^6^ copies/mL or the plasma BKPyV load was more than 2000 copies/mL, a kidney biopsy was conducted for histological diagnosis of BKPyVAN. Seventeen patients were diagnosed with biopsy-proven BKPyVAN based on histological findings according to the criteria of the 2018 Banff Working Group classification of definite polyomavirus nephropathy. The other 227 kidney recipients were classified into the non-BKPyVAN group.

The patients’ demographic characteristics, clinical manifestation, allograft survival and patient survival until the last visit were recorded. The eGFR calculated using the Modification of Diet in Renal Disease Study equation at last visit or before allograft failure was also collected for analysis. Acute rejection included acute T-cell-mediated rejection and acute antibody-mediated rejection. Mortality indicated all-cause mortality. 

### 2.2. Statistical Analysis

Continuous variables are expressed as the mean ± standard deviations, and categorical variables are expressed as a number (%). For comparisons among two patient groups, continuous variables were analyzed via Student’s *t*-test, whereas categorical variables were analyzed with a chi-square test or Fisher’s exact test. *p* values of < 0.05 were considered statistically significant. Allograft survival, urinary tract cancer-free survival rate and patient survival were determined by Kaplan–Meier analysis. Differences in survival curves between the two groups were analyzed by a log rank test. Univariate logistical regression analysis was used to compare the frequencies of selected risk factors for urinary tract cancers. To control for potential confounding factors and assess the risk factors used in the univariate analysis, a multivariate logistical regression analysis (enter method) was performed.

## 3. Results

### 3.1. Patient Characteristics

This observation cohort study included a total of 244 kidney transplant recipients in our hospital, and the mean of the follow-up period was 107 ± 83 months, with 25 (10.2%) recipients failing to follow-up. Among these subjects, 17 patients (7.0%) had biopsy-proven BKPyVAN and were classified into the BKPyVAN group. Two hundred and twenty-seven patients without BKPyVAN were classified into the non-BKPyVAN group. The demographics and laboratory measurements are summarized in Table 1. The mean age of the patients was 55.2 years old, and 65 subjects (26.6%) received living-donor transplantation. There was male predominance in the BKPyVAN group, but the difference of the sex ratio between two groups (70.6% vs. 49.3%) did not reach statistical significance. The BKPyVAN patients were younger than the non-BKPyVAN patients (48.4 ± 13 vs. 55.7 ± 12.9, *p* = 0.038). The average of the pre-transplant dialysis period was 4.3 years, and the pre-transplant dialysis period did not differ in the two groups. There was no difference in the prevalence of diabetes mellitus, HBV infection, and HCV infection. The patients in the BKPyVAN group had worse kidney function when compared to the patients in the non-BKPyVAN group (eGFR: 16 ± 15 vs. 41 ± 32 mL/min/1.73 m^2^).

### 3.2. The Worst Allograft Outcome in Patients with BKPyVAN

In this study, the mean age at the time of diagnosis of BKPyVAN was 41.5 ± 12.8 years old (23–60 years old) and the median time of kidney transplantation to the diagnosis of BKPyVAN was 16 months (1–97 months). Ten patients (59%) developed BKVAN within 1 year after transplantation and 15 patients (88%) within 2 years. During follow-up, more patients in the non BKPyVAN group experienced acute rejection episodes than those in the BKPyVAN group (*p* = 0.047). Although less acute rejection episodes were observed in the patients with BKPyVAN, the allograft outcome of these patients was worse than that of those without BKPyVAN (Figure 1A and Table 1). The 1-, 2-, and 5-year allograft survival for the patients with BKPyVAN was 100%, 81%, and 67%, respectively. The corresponding values for the non BKPyVAN patients were 99.5%, 97%, and 93%, respectively (*p* = 0.0002) (Figure 1A). There were eventually 9 (52.9%) allograft losses out of 17 patients with BKPyVAN, while there were 46 (20.3%) allograft losses out of 227 patients without BKPyVAN (*p* = 0.004) (Table 1).

Despite more allograft losses in the BKPyVAN group, the 10-year patient survival was not different in the two groups as the result of the log-rank test demonstrated no significant statistical difference between these two groups (*p* = 0.173) (Figure 1B). During follow-up, there were 3 deaths (17.6%) and 35 deaths (15.4%) in the BKPyVAN and non BKPyVAN groups, respectively, and the mortality rate between these groups was not significantly different.

### 3.3. Increased Incidence of Urinary Tract Cancers in Patients with BKPyVAN

Among 244 kidney transplant recipients, a total of 31 cancers (12.7%) were identified in this cohort study (Table 1 and Table 2). The incidence of all-type cancers in patients with BKPyVAN (29.4%) was significantly higher than that in patients without BKPyVAN (29.4% vs. 11.5%) (Table 1). Further analysis showed that the difference in cancer incidence between two groups was mainly attributed to significantly higher incidence of urinary tract cancers including RCC and UC in the BKPyVAN group compared to that in the non-BKPyVAN group (29.4% vs. 6.6%). There were no other cancers occurring in patients with BKPyVAN, while 11 (4.8%) non-BKPyVAN patients had cancers other than RCC and UC. However, this difference did not reach statistical significance.

We further assessed the difference in urinary tract cancer-free survival between two groups by Kaplan–Meier analysis. The urinary tract cancer-free survival rate in patients with BKPyVAN was significantly lower when compared to that in patients without BKPyVAN (Figure 2A). Since the longest observation period was 172 months in patients with BKPyVAN, we limited the comparison within 172 follow-up months. Similarly, the urinary tract cancer-free survival rate in patients with BKPyVAN was significantly lower than that in patients without BKPyVAN at the end of 172 observation months (Figure 2B). 

### 3.4. Earlier Onset of Urinary Tract Cancers in Patients with BKPyVAN

As the significant difference in the urinary tract, cancer-free survival between the BKPyVAN and non-BKPyVAN groups was observed in early post-transplant time, we further assessed whether the patients with BKPyVAN had early onset of urinary tract cancers. The mean onset of urinary tract cancers in patients with BKPyVAN was 59.8 months, while the mean onset of urinary tract cancers in patients without BKPyVAN was 108.9 months. However, this difference between the two groups did not reach statistical significance due to small case numbers (five urinary tract cancers in patients with BKPyVAN). We therefore enrolled another hospital, Taichung Veteran General Hospital, to reassess the onset of urinary tract cancers. Five out of twenty-eight patients with BKPyVAN in their hospital had urinary tract cancers (urothelial carcinoma of bladders) with a mean onset of urinary tract cancers, 47.0 months. Together with our patients, the mean onset of urinary tract cancers in these BKPyVAN patients was 53.4 months, which was significantly faster than the patients without BKPyVAN (*p* = 0.0259) (Figure 3). This result suggests earlier onset of urinary tract cancers in patients with BKPyVAN compared to patients without BKPyVAN.

### 3.5. Association of BKPyVAN and Urinary Tract Cancers

The risk factors of the urinary tract cancer development were determined by the logistic regression analysis (Table 3). Among the assessed factors including sex, age, diabetes, pre-transplant dialysis duration, transplant types, acute rejection, HBV, HCV infection, and BKPyVAN, the univariate logistic regression analysis identified that BKPyVAN was the only risk factor of urinary tract cancers. After adjusting the confounding factors, the multivariate logistic regression analysis demonstrated that age (RR = 1.062) and BKPyVAN (RR = 6.459) were the significant risk factors for the development of urinary tract cancers.

For cancers other than urinary tract cancers, the univariate logistic regression analysis showed that HBV infection was the only risk factor (Table 4). After adjusting the confounding factors, none of these factors were the risk factor for the development of non-urinary tract cancers. These results suggest that BKPyVAN was an important risk factor for the development of urinary tract cancers but not for other cancers.

## 4. Discussion

Since use of potent immunosuppressants gradually increased in the last two decades, BKPyVAN has become an emerging problem for kidney transplant recipients [35]. In the current study, 17 out of 244 (7.0%) kidney transplant recipients had the diagnosis of BKPyVAN and this prevalence was similar to the reported prevalence (1–10%) of other studies after year 2000 [34,36,37,38]. The median time of kidney transplantation to the diagnosis of BKPyVAN was 16 months, and 88% of BKPyVAN occurred within two years after transplantation. These findings were similar to previous reports [39,40]. Early development of BKPyVAN is observed in our and other studies, possibly reflecting overexpression of host immunity. This speculation was further strengthened by the finding in this study that acute rejection episodes occurring most in the first post-transplant year in patients with BKPyVAN were less than those in patients without BKPyVAN. The consequence of increasing potent immunosuppressants to prevent acute rejection is an early development of BKPyVAN, suggesting the requirement of regularly frequent surveillance for BKPyV reactivation in early post-transplant periods.

In line with other studies, more allograft losses were observed in our BKPyVAN group (Table 1). Furthermore, the 5-year allograft survival (67%) was significantly lower in patients with BKPyVAN compared to that (93%) in patients without BKPyVAN (Figure 1A). In the literature, the 5-year allograft survival in patients with BKPyVAN ranges from 30% to 70%, depending on the severity of BKPyVAN at diagnosis [38,39,40]. Although more allograft losses were observed in patients with BKPyVAN, the difference in 10-year patient survival between the BKPyVAN group and non-BKPyVAN group was not statistically significant. After allograft failure, the patients returned to dialysis and with artificial renal support the patient survival was not markedly reduced. This may explain the result of a significant difference in the 5-year graft survival but no difference in the 10-year patient survival.

Post-transplant malignancies are estimated about 20% after 10-year kidney transplantation and almost 30% after 20-year kidney transplantation [41,42]. It will be the next challenge as malignancies will be the major cause of death in kidney transplant recipients in the next 2 decades [24]. In the current study, the overall incidence of cancers in kidney transplant recipients from 2000 to 2020 was 12.7%, which is more than the reported incidence (9.5%) from 1996 to 2010 in the Taiwan nationwide, population-based study [43]. The difference may be attributed to the longer follow-up period in our study (20 years vs. 15 years). Nevertheless, in consistence with previous studies showing that urinary tract cancers are the most common post-transplant cancers in Taiwan, our study demonstrated that 20 out of 31 (65%) post-transplant cancers were urinary tract cancers [3,4,5]. The types of post-transplant cancers are different from those in western countries, where the most common cancers are skin cancers and lymphoma [2,44]. 

BKPyVAN development indicates overexpression of host immunity and impaired immune surveillance and clearance for malignant cells lead to increased cancer development. In the current study, the incidence of all cancers was significantly higher in the BKPyVAN group compared to the non-BKPyVAN group. In further cancer subtype analysis, this difference between two groups was mainly caused by higher proportion of urinary tract cancers in the BKPyVAN groups (BKPyVAN vs non-BKPyVAN 29.4% vs. 6.6%) and the incidence of other cancers was not significantly different (0% vs. 4.8%). This result was further verified by Kaplan–Meier analysis showing that the urinary tract cancer-free survival was significantly lower in patients with BKPyVAN when compared with patients without BKPyVAN (Figure 2A,B). These results suggest a strong association between BKPyVAN and urinary tract cancers.

In addition, the urinary tract cancer-free survival curves of the two groups were separated at early stage after kidney transplantation, suggesting an early onset of urinary tract cancers in patients with BKPyVAN. As expected, the mean onset of urinary tract cancers in patients with BKPyVAN was earlier than that in patients without BKPyVAN (59.8 vs. 108.9 months). Nevertheless, this difference did not reach statistical significance due to only five urinary tract cancers in BKPyVAN patients. We therefore enrolled 28 BKPyVAN patients in another tertiary teaching hospital, and among these patients, 5 patients had urothelial carcinoma of the bladder. The mean onset of post-transplant urinary tract cancers in these 10 BKPyVAN patients was 53.4 months, which was significantly earlier compared with that in non-BKPyVAN patients (Figure 3). These findings show not only increased incidence but also much earlier onset of urinary tract cancers in BKPyVAN patients, suggesting the oncogenic potential of BKPyV in the development of urinary tract cancers. BKPyV infection has been associated with urinary tract cancers in kidney transplant recipients in several case reports. A case-matched study demonstrated that among all cancers, only urothelial carcinoma of bladder was associated with BKPyV replication in kidney transplant recipients [34]. A nationwide, population-based study using the US transplant registry showed a strongly elevated incidence of invasive bladder cancer and borderline increased incidence of all urothelial cancers in kidney transplant recipients with BKPyVAN [3]. However, these studies did not compare the onset of urinary tract cancers in BKPyVAN and non-BKPyVAN patients. Our study showing a much earlier onset of urinary tract cancers in BKPyVAN patients adds more evidence to support the association between BKPyV and urinary tract cancers. 

Liu et al. demonstrated that BKPyV replication is a risk factor of bladder cancer with a high relative-risk value (RR = 11.7), even higher than those of smoking (RR = 6.1) and age (RR = 4.7) [34]. In accordance with their findings, our study also demonstrated that age and BKPyVAN were two independent risk factors associated with urinary tract cancers after adjusting the selected confounding factors by multivariate logistical regression analysis. Especially, BKPyVAN was a high relative-risk factor (BKPyVAN: RR = 6.459; age: RR = 1.062). These findings further confirm strong association between BKPyV infection and urinary tract cancers.

Although the causal relationship between BKPyV infection and the development of urinary tract cancers has not been identified in humans, Kenan et al. reported that the BKPyV genome integrated into the human genome in a urothelial carcinoma specimen and a renal cell carcinoma specimen, indicating the oncogenic potential of BKPyV [28,29]. Viral oncogenesis in the integrative viruses is the process of epigenetic and genetic alterations of the host cells during integration of the viral genome. During integration, viruses can reprogram the host’s cellular signaling pathways to stimulate pro-cancer inflammation and induce malignant transformation [45,46,47]. Kenan’s study may explain the oncogenic potential of BKPyV, as the integrated BKPyV genome can persistently produce oncogenic TAg expression to trigger malignant transformation of the host cells.

The major limitation of this study is its small case number for urinary tract cancers in BKPyVAN patients. Nevertheless, only 1–10% of kidney transplant recipients have BKPyVAN, and urinary tract cancers in BKPyVAN are reported with small case-number studies or case reports. Our study was conducted in an area with a high incidence of urinary tract cancers and supports the strong association between BKPyV infection and urinary tract cancers.

## 5. Conclusions

Our study demonstrates that despite similar patient survival, BKPyVAN patients have earlier and greater allograft losses compared to non-BKPyVAN patients. These BKPyVAN patients have high incidence, low cancer-free survival rates, and early onset, along with a high relative risk of developing urinary tract cancers.

## Figures and Tables

**Figure 1 viruses-13-00476-f001:**
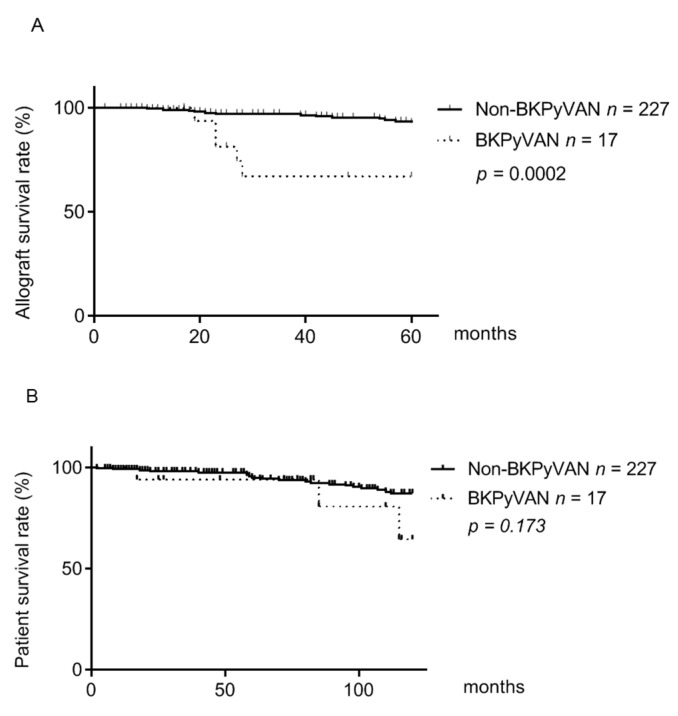
Kaplan–Meier analysis of allograft survival and patient survival in patients with and without BKPyVAN. Patients were classified into the BKPyVAN and non-BKPyVAN groups. The allograft survival (**A**) and patient survival (**B**) were assessed by Kaplan–Meier analysis.

**Figure 2 viruses-13-00476-f002:**
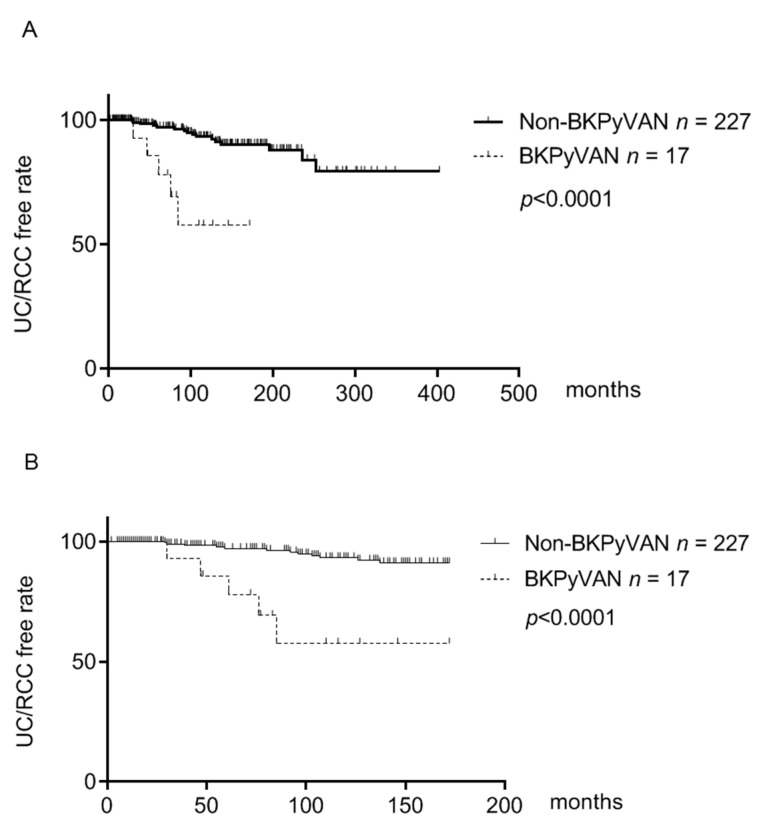
Kaplan–Meier analysis of urinary tract cancer-free survival in patients with and without BKVAN. (**A**) Urinary tract cancer (urothelial carcinoma (UC) and renal cell carcinoma (RCC))-free survival in patients with and without BKPyVAN and non-BKPyVAN was determined by Kaplan–Meier analysis. (**B**) As the longest observation period was 172 months in patients with BKPyVAN, urinary tract cancer-free survival within 172 months in two groups was assessed by Kaplan–Meier analysis.

**Figure 3 viruses-13-00476-f003:**
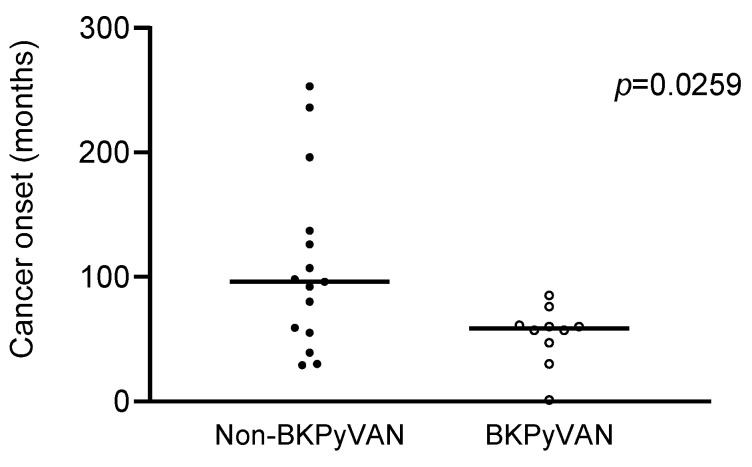
Comparison of the onset of urinary tract cancers after kidney transplantation between patients with and without BKPyVAN. The onset of 10 urinary tract cancers (5 from our hospital and 5 from Taichung Veteran General Hospital) after kidney transplantation in patients with BKPyVAN was compared to that in patients without BKPyVAN by Student’s *t*-test.

**Table 1 viruses-13-00476-t001:** Clinical characteristics of kidney transplant recipients with/without BKPyVAN.

	Total *n* = 244	BKPyVAN *n* = 17	Non-BKPyVAN *n* = 227	*p* Value
Gender (male)	124 (50.8%)	12 (70.6%)	112 (49.3%)	0.13
Age (years old)	55.2 ± 13.1	48.4 ± 13.0	55.7 ± 12.9	0.038
Diabetes	65 (26.6%)	4 (23.5%)	61 (26.9%)	1.0
Living related donor	65 (26.6%)	3 (17.6%)	62 (27.3%)	0.57
Dialysis duration (year)	4.3 ± 4.3	5.8 ± 6.0	4.1 ± 4.1	0.284
Hepatitis virus B	21 (8.6%)	0 (0%)	21 (9.3%)	0.38
Hepatitis virus C	31 (12.7%)	2 (11.8%)	29 (12.8%)	1.0
Creatinine (mg/dL)	3.3 ± 3.1	5.7 ± 3.1	3.1 ± 3.0	0.002
eGFR (mL/min/1.73 m^2^)	40 ± 32	16 ± 15	41 ± 32	<0.0001
Acute rejection episode	67 (27.5%)	1 (5.9%)	66 (29.1%)	0.047
Allograft loss	55 (22.5%)	9 (52.9%)	46 (20.3%)	0.004
Urinary tract cancers ^a^	20 (8.2%)	5 (29.4%)	15 (6.6%)	0.007
Cancers other than urinary tract cancers	11 (4.5%)	0 (0%)	11 (4.8%)	1.0
Any cancer	31 (12.7%)	5 (29.4%)	26 (11.5%)	0.048
Mortality ^b^	38 (15.6%)	3 (17.6%)	35 (15.4%)	0.734

a: Total: 17 urothelial carcinomas (UC) and 3 renal cell carcinomas (RCC); BKPyVAN group: 4 UC and 1 RCC; non BKPyVAN group: 13 UC and 2 RCC. b: Mortality: all-cause mortality.

**Table 2 viruses-13-00476-t002:** Subtypes of cancers in kidney transplant recipients with/without BKPyVAN.

	Total *n* = 244	BKPyVAN *n* = 17	Non-BKPyVAN *n* = 227
Urinary tract cancer			
Renal cell carcinoma	3	1	2
Pelvis/ureter	1	0	1
Bladder	16	4	12
Prostate	0	0	0
Total	20 (8.2%)	5 (29.4%)	15 (6.6%)
Other cancers			
Hepatocellular carcinoma	2	0	2
Lymphoma	2	0	2
Cervical cancer	2	0	2
Endometrial cancer	1	0	1
Breast cancer	1	0	1
Colon cancer	1	0	1
Lung cancer	1	0	1
Kaposi sarcoma	1	0	1
Total	11 (4.5%)	0 (0%)	11 (4.8%)

**Table 3 viruses-13-00476-t003:** Univariate and multivariate logistic regression analysis to identify relative risks of urinary tract cancers in kidney transplant recipients.

Factor	RR	Hazard Ratio (95% CI)	*p* Value
Univariate logistic regression			
Sex (male)	0.775	0.309–1.044	0.588
Age	1.036	0.995–1.079	0.085
Living donor	0.668	0.215–2.077	0.486
Pre-transplant dialysis duration	1.049	0.945–1.164	0.369
Diabetes	0.461	0.131–1.628	0.229
HBV	1.000		0.998
HCV	1.823	0.568–5.860	0.313
eGFR	0.980	0.961–1.000	0.053
Acute rejection	0.441	0.125–1.557	0.203
BKPyVAN	5.889	1.833–18.923	0.003
Multivariate logistic regression			
Age	1.062	1.006–1.122	0.030
BKPyVAN	6.459	1.131–36.889	0.036

**Table 4 viruses-13-00476-t004:** Univariate and multivariate logistic regression analysis to identify relative risks of urinary tract cancers in kidney transplant recipients.

Factor	OR	Hazard Ratio (95% CI)	*p* Value
Univariate logistic regression			
Sex (male)	0.798	0.237–2.689	0.716
Age	1.030	0.977–1.085	0.269
Living donor	0.000		0.997
Pre-transplant dialysis duration	1.048	0.906–1.213	0.529
Diabetes	0.264	0.033–2.105	0.209
HBV	4.479	1.092–18.372	0.037
HCV	0.677	0.084–5.477	0.714
eGFR	0.993	0.973–1.015	0.539
Acute rejection	0.990	0.255–3.849	0.989
BKPyVAN	0.000	1.833–18.923	0.999
Multivariate logistic regression			
HBV	5.205	0.782–34.649	0.088

## Data Availability

Our data is not available to the public.

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
