# Peer review of "High Incidence and Early Onset of Urinary Tract Cancers in Patients with BK Polyomavirus Associated Nephropathy"

_viruses, 2021, doi:10.3390/v13030476_

Round 1
Reviewer 1 Report
Broad comments
The subject of the article "High incidence and early onset of urinary tract cancers in patients with polyomavirus BK-associated nephropathy" fits the aims and scope of Viruses. The subject of the study is important for clinicians and shows important dependencies.
All sections: Introduction, Material and methods, Results and Discussion are well and interestingly written. The Introduction, although short, is well written and introduces the reader to the topic. Study included 244 kidney transplant recipients in a 20- year cohort - so the research material is representative. Both the description of the patients and the statistical methods used to develop the results seem correct. The Results are clearly described, as are the accompanying graphs and tables.
The discussion is quite short, but this is largely due to the innovation of the research presented and the lack of literature for discussion.
Specific comments
Abstract
Page 1: Line: 16 "Background Over-immunosuppressed kidney transplant recipients are susceptible to..." - delete "Background"
Please standardize the number format (sometimes there is a space, sometimes there is no)
Please standardize the number format - there is a different number of decimal places in different numbers.
Page:2 Line: 88 0 remove red color
Page: 10 Line: 331 - remove red color
Author Response
Responding to the opinions of the reviewer 1.
We would like to thank the reviewer for the recommendations to improve this manuscript.
Broad comments
The subject of the article "High incidence and early onset of urinary tract cancers in patients with polyomavirus BK-associated nephropathy" fits the aims and scope of Viruses. The subject of the study is important for clinicians and shows important dependencies. All sections: Introduction, Material and methods, Results and Discussion are well and interestingly written. The Introduction, although short, is well written and introduces the reader to the topic. Study included 244 kidney transplant recipients in a 20- year cohort - so the research material is representative. Both the description of the patients and the statistical methods used to develop the results seem correct. The Results are clearly described, as are the accompanying graphs and tables. The discussion is quite short, but this is largely due to the innovation of the research presented and the lack of literature for discussion.
A: We would like thank the reviewer for the comments and appreciate the reviewer’s understanding of the limitations of this study.
Specific comments
Abstract
- Page 1: Line: 16 "Background Over-immunosuppressed kidney transplant recipients are susceptible to..." - delete "Background"
A: We thank the reviewer for pointing out our error. We have deleted “Background” in the abstract section.
- Please standardize the number format (sometimes there is a space, sometimes there is no)
A: We thank the reviewer for pointing out our errors. We have corrected the errors and used the same number format (one space between the numbers) throughout the text.
- Please standardize the number format - there is a different number of decimal places in different numbers.
A: We thank the reviewer for the recommendation. We have amended our errors as suggested.
- Page:2 Line: 88 0 remove red color
A: We thank the reviewer to remind us the mistake. We have changed the red-color word to the black-ink word.
- Page: 10 Line: 331 - remove red color

Reviewer 2 Report
GENERAL COMMENT
The work "High incidence and early onset of urinary tract cancers in patients with polyomavirus BK-associated nephropathy" by Li et al. reports on the association between BKPyV infection and urinary tract cancers (UTC). Authors demonstrates that BKPyVAN patients have greater allograft losses, higher incidence, a lower cancer-free survival rate, and an earlier onset with a higher relative risk of developing UTC compared to non-BKPyVAN patients. The limit of this observational study is few cases reported.
SUGGESTIONS FOR AUTHORS
- Since 2016 the taxonomy/nomenclature of polyomaviruses has been updated/changed, including BKV (BK virus), which is now BKPyV (BK polyomavirus). Please see Archives ofVirology volume 161, pages1739–1750(2016). Authors shoud use the new BKPyV in their revised text.
- In the Introduction section authors state, line 54-55: Among these polyomaviruses, SV40 seems not to exist in human cancers. This sentence is not correct. A better sentence could be: Among these polyomaviruses, the role of SV40 in the onset of human cancers remains to be clarified. Indeed, many investigations reported the association between SV40 and human cancers, while other studies publhished negative data. A recent review has been published in this field, such as doi.org/10.3389/fonc.2019.00670
-
This sentence "However, the direct causal role of these viruses in tumorigenesis remains inconclusive" should be re-written, such as "However, the direct causal role of these small DNA tumor viruses in human cancers needs to be investigated further with new innovative methods".
- This sentence "An epidemiologic study demonstrated that 80-100% of the general population was infected by BKV in their childhood [15]" is based on a old reference. More recent articles, which should be cited, reported a lower prevalence in the general population and children. Similarly, the citation 24 is very old. New additional studies/reviews have been published since 1982!
- In general, too many old studies have been reported in the text, introduction , discussion, reference sections. More recent investigations should be cited, whereas new references added instead of the old references. Accordingly, the revised text should enclose major revisons.
Author Response
Responding to the opinions of the reviewer 2.We would like to thank the reviewer for the constructive recommendations. We have amended the manuscript as suggested. The changed text is highlighted in red color.
GENERAL COMMENT
The work "High incidence and early onset of urinary tract cancers in patients with polyomavirus BK-associated nephropathy" by Li et al. reports on the association between BKPyV infection and urinary tract cancers (UTC). Authors demonstrates that BKPyVAN patients have greater allograft losses, higher incidence, a lower cancer-free survival rate, and an earlier onset with a higher relative risk of developing UTC compared to non-BKPyVAN patients. The limit of this observational study is few cases reported.
A: We agree with the reviewer about small case number of BKPyVAN in this study. It would be better to conduct multi-center analysis as BKPyVAN is relatively rare in most transplant centers. This study provides an important observation and multi-center studies will be required to verify the finding of this study. The collaborative project of multi-transplant centers is undergoing.
SUGGESTIONS FOR AUTHORS
- Since 2016 the taxonomy/nomenclature of polyomaviruses has been updated/changed, including BKV (BK virus), which is now BKPyV (BK polyomavirus). Please see Archives of Virologyvolume 161, pages1739–1750(2016). Authors should use the new BKPyV in their revised text.
- In the Introduction section authors state, line 54-55: Among these polyomaviruses, SV40 seems not to exist in human cancers. This sentence is not correct. A better sentence could be: Among these polyomaviruses, the role of SV40 in the onset of human cancers remains to be clarified. Indeed, many investigations reported the association between SV40 and human cancers, while other studies published negative data. A recent review has been published in this field, such as doi.org/10.3389/fonc.2019.00670
- This sentence "However, the direct causal role of these viruses in tumorigenesis remains inconclusive" should be re-written, such as "However, the direct causal role of these small DNA tumor viruses in human cancers needs to be investigated further with new innovative methods".
- This sentence "An epidemiologic study demonstrated that 80-100% of the general population was infected by BKV in their childhood [15]" is based on an old reference. More recent articles, which should be cited, reported a lower prevalence in the general population and children. Similarly, the citation 24 is very old. New additional studies/reviews have been published since 1982.
- In general, too many old studies have been reported in the text, introduction, discussion, reference sections. More recent investigations should be cited, whereas new references added instead of the old references. Accordingly, the revised text should enclose major revisions.

Round 2
Reviewer 2 Report
Please modify JCV with JCPyV